# Bacterial Biosorbents, an Efficient Heavy Metals Green Clean-Up Strategy: Prospects, Challenges, and Opportunities

**DOI:** 10.3390/microorganisms10030610

**Published:** 2022-03-13

**Authors:** Van Hong Thi Pham, Jaisoo Kim, Soonwoong Chang, Woojin Chung

**Affiliations:** 1Department of Environmental Energy Engineering, Graduate School of Kyonggi University, Suwon 16227, Korea; vanhtpham@gmail.com; 2Department of Life Science, College of Natural Science of Kyonggi University, Suwon 16227, Korea; jkimtamu@kyonggi.ac.kr; 3Department of Environmental Energy Engineering, College of Creative Engineering of Kyonggi University, Suwon 16227, Korea

**Keywords:** heavy metal removal, biosorbent, bacterial biomass, microbial metal removing strategy

## Abstract

Rapid industrialization has led to the pollution of soil and water by various types of contaminants. Heavy metals (HMs) are considered the most reactive toxic contaminants, even at low concentrations, which cause health problems through accumulation in the food chain and water. Remediation using conventional methods, including physical and chemical techniques, is a costly treatment process and generates toxic by-products, which may negatively affect the surrounding environment. Therefore, biosorption has attracted significant research interest in the recent decades. In contrast to existing methods, bacterial biomass offers a potential alternative for recovering toxic/persistent HMs from the environment through different mechanisms for metal ion uptake. This review provides an outlook of the advantages and disadvantages of the current bioremediation technologies and describes bacterial groups, especially extremophiles with biosorbent potential for heavy metal removal with relevant examples and perspectives.

## 1. Introduction

The rapidly escalating industrial activities release toxic heavy metals (HMs), which pose a serious hazard to ecosystems and human health [1,2,3]. Environmental HM pollution in soil and water reduces crop production and can be detrimental to health safety through food chains owing to industrial solid waste, and agricultural inputs such as fertilizers and pesticides. These persistent environmental contaminants are non-degradable and can only be transformed into other harmless forms, such as Hg, Cd, As, Cr, Tl, Pb, Mn, and Ni, which cause severe toxic effects in living organisms [4,5,6,7,8]. Fe, Cu, Co, and Zn are essential HMs that act as coenzymes in biological processes and are less toxic at low concentrations [9]. Over the last few decades, many conventional treatment methods have been used to remove HMs from polluted environments, including chemical precipitation, ultrafiltration, ion exchange, reverse osmosis, electrowinning, and phytoremediation [10]. The traditional methods used are described in Table 1.

Therefore, the development of remediation treatment methods is essential for mitigating the negative effects on nature. Because of the drawbacks of conventional remediation methods, such as high cost and lack of environmentally friendly solutions, green technologies are growing in the investigation of biosorbents and potential microbial biomass. This is because of their high removal/recovery efficiency, low cost, and safety to restore polluted environments [11]. However, in such cases, the speed of pollutants released by bacteria is usually low and has a short lifespan because of dead biomass, which limits their feasibility in large-scale applications [12,13]. As an alternative, numerous studies have confirmed that using enzymes and bio-surfactants produced from microbes is more advantageous than using microbes as a whole to boost remediation efficacy as biocatalysts [14,15]. Enzymatic degradation requires highly specific and flexible operating conditions. Therefore, exploring enzyme-producing bacterial sources remains attractive. For instance, ChrR and YieF are two soluble enzymes that have been extracted and purified from *Pseudomonas putida* MK1 and *Escherichia coli*, respectively; these are capable of effectively reducing Cr^6+^ to Cr^3+^ under both aerobic and anaerobic conditions [14].

Recently, the use of beneficial microorganisms, such as plant growth-promoting bacteria, which are also capable of reducing HMs, has significantly contributed to agriculture and environmental schemes owing to its outstanding advantages. Numerous reports have demonstrated that several bacteria can adapt to high levels of heavy metal pollution [16,17,18]. Bacteria use chemical contaminants as an energy source through their metabolic processes; however, excessive amounts of inorganic nutrients pose a risk to their metabolism [19,20,21]. Bacterial groups that contribute to HM removal include *Bacillus* sp., *Pseudomonas* sp., *Arthrobacter* sp., *Alcaligenes* sp., *Azotobacter* sp., *Rhodococcus* sp., and methanogens [22]. Among them, *Bacillus* sp. is considered a potential agent for removing various HMs, especially Gram-positive bacteria [23]. Oves et al. investigated *Bacillus thuringiensis* OSM269 that was tolerant to various concentrations (25–150 mg/L) of HMs, such as Cd, Cr, Cu, Pb, and Ni [24]. Moreover, because of their diverse enzymatic systems, members of the *Streptomyces* genus have recently been evaluated as important producers in the remediation of contaminated environments [25]. In a previous study, two *Pseudomonas* strains were shown to be resistant to As and other HMs such as Ag, Cd, Co, Cr, Cu, Hg, Ni, and Pb [26]. The biosorption of Al^3+^ and Cd^2+^ by an extra cellular polymeric substance (EPS) from *Lactobacillus rhamnosus* was determined in a previous study [27]. In another study, one member of the genus *Cellulosimicrobium* was found to be a potential bacterium that can protect against six HMs, including Pb, Fe, Cd, Ni, Cu, and Co [28]. Recently, an exopolysaccharide produced by *Lactiplantibacillus plantarum* BGAN8 strain was discovered to have a high Cd-binding capacity and prevent cadmium-induced toxicity [29].

Thus, this review addresses the importance of the roles played by bacteria in both biosorption and bioaccumulation platforms in HM recovery. In-depth studies on biological phenomena are required to understand the mechanisms by which microbes use proteins to uptake HMs in the intracellular space. This review also highlights the advantages, drawbacks, obstacles, and potential avenues of promising unculturable bacteria, especially extremophiles for research and practical application in HMs removal.

## 2. Microbial Remediation: The Mechanisms of Biosorption and Bioaccumulation Using Bacterial Biomass as a Tool in Polluted Environmental Cleanup

The metabolic diversity and activity of microorganism have provided tremendous potential in the field of waste treatment via cell owners of various metabolic pathways (Figure 1). Toxic compounds have been used as energy sources for cellular processes through fermentation, respiration, and co-metabolism [30].

Owing to the various considerations of each group, as well as under certain experimental conditions, various microbial biomasses have different bioremediation abilities. HMs may disrupt microbial cell membranes, but bacteria possess characteristic enzymatic profiles that are required to overcome toxic effects. The bioremediation process takes place through various mechanisms, including:Alkylation and redox processes in which HMs were transformed. The speciation and mobility of metal (loids) may be different from the initial state. For example, metals generally are less soluble in their oxidation state, whereas the solubility and mobility of metalloids depend on both the oxidation state and the ionic form [31].Passive adsorption is metabolism-independent, in which metals are on the cell surface via electrostatic attraction with functional groups. This mechanism was explained by the different processes including the precipitation and the surface complexation, ion exchange as only dominating role, or physical adsorption. As protons were addressed by the completion between pH and metal cations on the binding sites, thus, pH is the most strongly effective factor that influences the biosorption process [32]. The other essential factors include temperature, ionic strength, the concentration and type of the sorbate and biosorbent, the state of biomass: suspension or immobilized and the presence of other anion and cations in the growth medium. Most applications focus on the utilization of dead biomass because the toxicity of bacteria is avoided, no requirement for maintenance, and the storage of biomass is easy and can be kept for long period without loss of effectiveness. Numerous bacterial strains were reported in HMs biosorption that are dominant in *Bacillus, Pseudomonas, Streptomyces* [30,33,34].Active adsorption is the metabolism-dependent intracellular accumulation of toxicants in living cells within cytoplasm. HMs were converted to non-bioavailable form by binding with metallothioneins (MTs) as low-molecular mass cystein-rich proteins, and metallo-chaperones. By being bound with HMs, these intracellular proteins can also lower the free ion concentrations within cytoplasm in which the detoxification of metals occurred [35]. This process is sensitive to environmental conditions depending on each type of bacterial strain such as pH, temperature, salinity. Moreover, it also depends on the biochemical structure, physiological/genetic adaptation, and the toxicity of metal. *Cyanobacteria*, *pseudomonads*, and *mycobacteria* have been found as the candidates that can synthesize MTs. MTs are usually associated with Zn, Cu, and other toxic metals such as Cd, Hg, and Pb [36]. *Pseudomonas aeruginosa and Pseudomonas putida* were reported as MTs producing bacteria exposed to Ca and Cu contamination [37].The metal ions uptake is carried out by a complex mechanism of releasing EPS, such as proteins, DNA, RNA, and polysaccharides the slippery layer on the outside of the cell wall. These have a crucial role of stopping the penetration of metals into the intracellular environment in which, ion exchange may occur. Numerous bacterial strains were investigated for the commercial production of EPS such as *Stenotrophomonas maltophilia*, *Azotobacter chroococcum*, *Bacillus cereus* KMS3-1 [38,39,40]. Bioremediation efficiency by this mechanism relies on the type and amount of carbon source available and other abiotic stress factors like pH, temperature, and the growth phase of each bacterium [41].

### 2.1. Biosorption Process

Although bioaccumulation and biosorption are used synonymously and naturally, they differ in the ways they sequester contaminants. Volesky defined biosorption as the adsorption of substances from solution by biological materials using physiochemical pathways of uptake, such as electrostatic forces and ion/proton displacement [42]. This is based on ionic interactions between the extracellular surface of the dead biomass, living cells, and metal ions. Thus, the amount of contaminants binds to the surface of the cellular structure rather than oxidation through aerobic or anaerobic metabolism. Biosorption has been shown to effectively remove a variety of HMs from aqueous solutions, including highly toxic metal ions, such as Cd, Cr, Pb, Hg, and As [43,44]. The functional groups in the cell walls of bacteria are responsible for binding tasks, including carboxyl, phosphonate, amine, and hydroxyl groups [45]. Therefore, the success of biosorption relies on the diversity of cell wall structures. Gram-positive bacteria have been shown to contain a high sorption capacity because of their thicker peptidoglycan layer [46,47]. Recently, some studies have investigated engineered microorganisms in combination with metal-binding proteins and peptides on the extracellular surface to improve the capacity and specificity of microbial sorbents [20,48]. Biosorption can remove contaminants using microorganisms (live/dead), agriculture, and other industrial byproducts as biosorbents, providing a fundamental background for sustainable biosorption technology for metal removal and recovery [49,50]. The effect of several factors such as pH, temperature, shaking speed, initial concentration of pollutants or amount of biosorbent is evaluated to optimize the biosorption efficiency. The binding mechanism depends on the chemical nature of each contaminant, size of the biomass, interaction between different metallic ions, and ionic strength [51]. Additionally, biosorption is attractive owing to a number of advantages, including the simple requirement for operation, no additional nutrients, operating cost-effectiveness owing to its reversible process, no increase in the chemical oxygen demand (COD), desorption ease, and high adsorption rate. However, it is necessary to consider other factors such as the possible toxicity of the pollutants to bacterial cell in case of using living cell in this process. Table 2 reports the potential bacterial candidates that are capable of HMs removal by biosorption process. 

### 2.2. Bioaccumulation Process

However, bioaccumulation is a natural active metabolic process in which HMs accumulate and are taken up into intracellular living bacterial cells using proteins. Bioaccumulation occurs when the absorption rate of contaminants exceeds the rate of loss. This process requires respiration with energy into the cytoplasm through the cell metabolic cycle and occurs in two steps: the first step is considered to be the adsorption of HMs onto the cells and the metal species then transported inside the cells in which the HMs can be sequestered by proteins, the lipid bilayer as an import system, and peptide ligands as a storage system [84]. Therefore, this process depends on bacterial cell metabolism. 

In intracellular sequestration, metal ions to form large ion were uptake by several compounds inside cytoplasm of cell. In previous study, *Pseudomonas putida* shows the potential of intracellular sequestration of metal ions such as zinc, cadmium, and copper [85]. In extracellular sequestration of Gram-negative bacteria, the improvement of HMs removal relies on improving the uptake from the periplasm into the cytoplasm of bacterial cell through expression of inner membrane importers [86,87]. The bacterial strains investigated as the promising HMs removing candidates via bioaccumulation mechanism are listed in Table 3.

### 2.3. Difference in Attractive Spots of Biosorption and Bioaccumulation Process

Compared to biosorption where dead bacterial cells still are able to remove HMs, bioaccumulation only occurs with living bacterial cell. The different points of each process were described in Table 3. Solutions for biosorption were designed based on the conventional sorption methods by testing the microbes ability with attractive adsorption properties and investigating the mechanism of chemical modification on the outer surface of cells and in metal-binding proteins and peptides [20,48]. Therefore, the chemical structure of the cell wall plays an important role in the biosorption mechanism with the specific functional groups different from each type of microbe.

However, bioaccumulation is a more complex process that is concerned with the inner cell structure and space, the genetic features and cellular processes for enzyme catalysis with the tolerance of bacterial cell under harsh environmental conditions including toxic pollutants. Moreover, the efficiency of accumulation of a bacterial strain also depends on the case where the bacterial strain was isolated from, in natural areas with extreme conditions or growing with an adaptation in contaminated site. Even though, bioaccumulation has been studied for over two decades for the application of remediation but possibility of translation to industrial-scale is still limited. Therefore, bioaccumulation is more attractive with numerous open questions for identifying gaps in knowledge of researchers and potential values for bioextractive applications. 

Table 4 and Table 5 show the different points of biosorption and bioaccumulation and the advantages and disadvantages of bacterial biosorbent, respectively.

The sensitivity and capacity of organisms to uptake chemicals are highly variable, and rely on environmental factors such as temperature, pH, and moisture, which can affect the transformation and transportation, as well as the types of chemicals formed, redox potential, nutrient status, and the organism itself [84,102]. The factors affecting biosorption and bioaccumulation are shown in Figure 2. 

A change in the morphology and physiology of the cell was observed upon increasing the concentration of metal ions for accumulation. Additionally, organisms capable of accumulating HMs may tolerate one or more metals at higher concentrations [103]. As a result, toxic metal ions are detoxified or transformed into non-toxic, stable, and inert forms [104].

Sulfate-reducing bacteria (SRB) have been investigated as bio-tools for removing HMs from acid mines because they contribute to the formation of metal sulfides as toxic metals through sulfide reactions after the conversion of sulfate to sulfide [105]. The dominant microbial groups in the acid mine belonged to the phyla *Acidobacteria, Actinobacteria, Bacteroidetes, Firmicutes, Nitrospirae*, and some classes of phylum *Proteobacteria*. SRB are chemolithotrophic/chemoorganotrophic organisms that can utilize sulfate as the terminal electron acceptor [106]. Kefeni et al. (2017) determined the optimal conditions for acid mine tailings waste using mixtures of salts and available organic substrates such as manure, sawdust, mushroom compost, sugarcane waste, and wood chips as the carbon sources, and yeast extract as the nitrogen source for bacterial metabolism. Hydrogen sulfide gas reacts with HMs to form insoluble metal sulfide precipitates, which remove the metals [107].

## 3. Potential of Extremophiles in Heavy Metal Removing

Extreme environments are mostly habitats with extreme natural conditions, such as certain areas of the deep sea, volcanoes, and deserts with harsh temperatures, high salinity, and alkaline/acidic pH. However, recently these extreme stressful conditions have been reported more in our anthropogenic environments caused by extremely recalcitrant pollutants. Extremophilic bacteria are seen as attractive as they generally have special well-developed mechanisms for tolerating and removing heavy metals as well as in physical and geochemical extreme environments. They are not only extremely tolerant but can also detoxify toxic pollutants under adverse conditions through special cellular metabolism [108,109,110]. They can express defense mechanisms active against multiple extremes simultaneously [111,112]. They synthesize extremophilic enzymes and biomolecules that protect their survival and keep active or stable under severe stress. Indigenous species isolated from contaminated sites have been reported to demonstrate exceptional resistance and biosorption efficiency toward HMs. Siderophores are small biomolecules for metal scavenging, which are unusual structural and functional properties synthesized by extremophiles that also have been proved [113]. It is evident that extremophiles have potential as bioremediation agents for metal chelation. Additionally, extremophiles have fast adapting transcriptional and translational mechanisms that involve the metal detoxification pathways [114].

Recently, to address the problem of wastewater containing heavy metal under harsh environmental conditions, the removal of heavy metal bacterial strains has been investigated [115,116]. Biosorption of HMs, including Cd^2+^, Cu^2+^, Co^2+^, and Mn^2+^, was conducted at high temperatures of up to 80 °C using the thermophilic bacteria *Geobacillus thermantarcticus* and *Anoxybacillus amylolyticus* [117]. A Pb-resistant psychrotrophic bacterial strain has been found to serve as a biosorbent for Pb^2+^ at 15 °C [73]. Masaki et al. demonstrated the bioreduction and immobilization of Cr using the extremely acidophilic Fe (III)-reducing bacterium, *Acidocella aromatica* strain PFBC [118]. To enhance the stability of Cd turnover, Cd nanoparticles were provided to precipitate Cd using *Acidithiobacillus* spp. [119] and *Acidocella aromatica*. In a previous study, Cd, Cu, Zn, and Ni were removed from the acidic solution by a potential thermophilic *acidophilus*, more specifically, *Sulfobacillus thermosulfidooxidans* [120]. U (VI) and Fe^2+^ from contaminated mine water were removed using *Acidothiobacillus ferrooxidans* strains [121,122] Additionally, under extremely acidic conditions, acidophilic microorganisms have been used as host strains for detoxification of HMs, such as bioleaching and bio-oxidation [123,124,125]. The characteristics of each extremophile type are illustrated in Figure 3 and Figure 4 describing the mechanism of each type of extremophile.

Despite the great biotechnological potential of extremophiles, remediation efficiency, biomass productivity, and economic profitability, their application in situ is still challenging due to the complex adaption mechanism in extreme conditions, interspecies competition and interaction/communication inhibiting bioremediation, and limited proliferation [109,126]. Survival mechanisms of extremophilic microorganisms are still required to develop sustainable bioremediation processes because of the complex relationships between inside and outside bacteria and growth environments.

## 4. Future Prospects

Owing to their high diversity in the living world, bacteria properties provide a valuable source for bioremediation of multiple pollutants. There are enormous possible materials (living or dead biomass) that can be continuously investigated to find out the Aefficiency of HMs bioremediation. It has been proved that biosorption generally provides a greater sorption capacity in comparison with bioaccumulation [127]. The microbial genetic engineering has been developed to enhance the capacity of microorganisms to tolerate and accumulate HMs [128,129]. Moreover, immobilizations of bacterial biomass on suitable carrier also may be addressed to improve the porosity, physical, and chemical stability [130]. However, biosorption with immobilized bacterial biomass indicates drawbacks due to the exact mechanism of the process is still not totally understood [131]. In particular, extremophiles have been receiving special interest because they enclose a pool of genetic and metabolic opportunities for specific purpose that can be harnessed as microfactories for the removal of HMs as well as various types of pollutants. Nevertheless, the natural physiological features of such extremophilic microfactories can be further explored to nourish different devices of HMs bioremediation, which are figured out in the literature but have not been identified or integrated so far. Additionally, the application in situ of extremophiles is still challenging due to a limited proliferation and their interaction in community under stresses. The complexity of extremophiles in cellular mechanisms and interspecies relationships under complex co-occurring extreme conditions may increase their stress to sustainably enhance heavy metal bioremediation. In combination with conventional methods, current technologies still need much more effort to overcome the limitations in term of bacterial its-self metabolism, the complex interaction between microorganisms and between microorganisms with living environmental factors, such as, (1) the production of organic acid of each type of bacteria for pH calibration in the environment; (2) organic solutes in the water to compensate for the external osmotic pressure; (3) polysaccharides presence in the polluted sites which are constituents of biofilm; and (4) concentration of toxic pollutants. In such a bio-green approach, global changes can be maintained using available natural resources to meet the cost-efficiency requirements of the treatment process. Moreover, in combination with biotechnology and nanotechnology, a variety of other approaches exist that could provide advanced tools in bioengineering to improve the possibility of extremophiles for the treatment of environmental toxic pollutants.

## 5. Concluding Remarks

In this regard, a diversity of bacterial strains in various environments proved as a huge source with an important potential for heavy metals removal in which the complex and diverse mechanisms were involved at extracellular and intracellular level. However, the feasibility of the bioremediation process at large scale is still not fully demonstrated yet, considering both environmental and economic aspect. There is a need for a thorough investigation of the relationships and interactions between microbes and other communities in these habitats in the real polluted environments. During the bioremediation process, the challenges associated with each specific situation of environmental condition and eco-physiology should be considered and controlled from the starting phase to the end of the process. On the other hand, this would facilitate the development of new techniques for the isolation of multifunctional bacterial candidates capable of utilizing a large number of pollutants as nutrients for their growth.

## Figures and Tables

**Figure 1 microorganisms-10-00610-f001:**
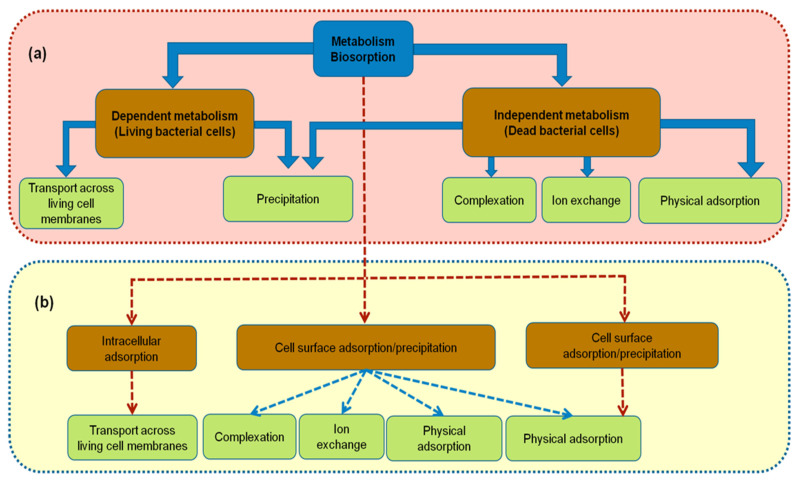
Mechanisms of bacterial biosorbent: (**a**) on cell surface; (**b**) within the cell where the HMs are removed.

**Figure 2 microorganisms-10-00610-f002:**
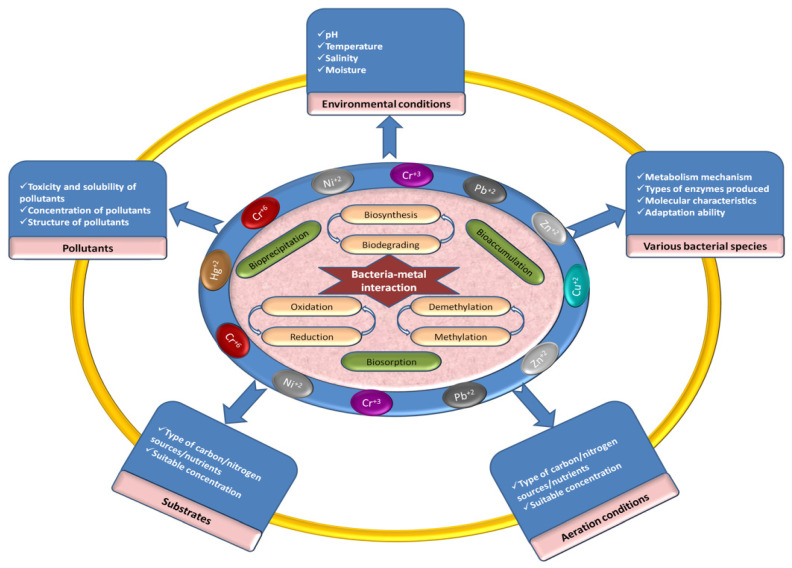
Factors affecting the bioremediation process.

**Figure 3 microorganisms-10-00610-f003:**
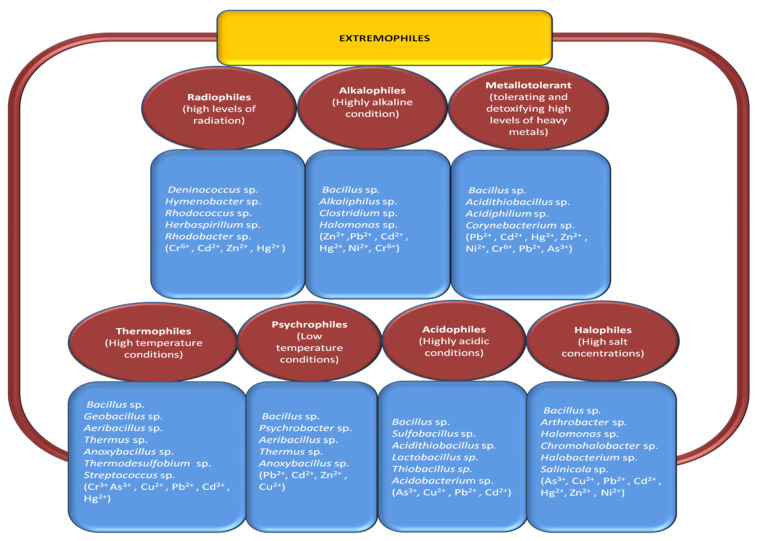
A schematic diagram representing extremophile candidates tolerant to different harsh environmental conditions that are capable of heavy metal removal.

**Figure 4 microorganisms-10-00610-f004:**
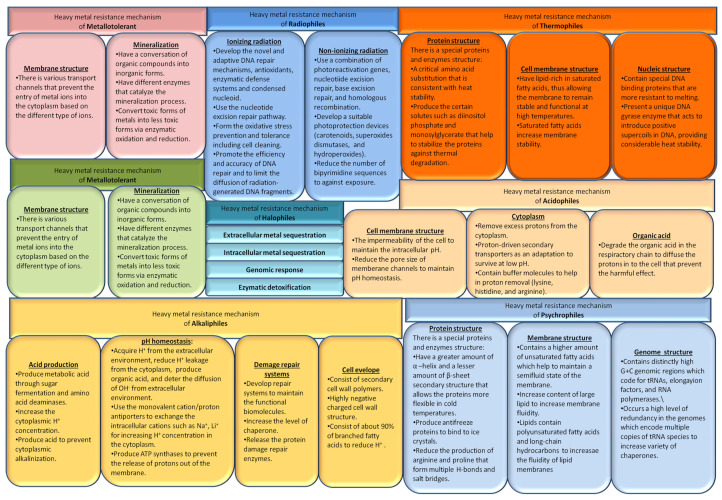
Mechanisms of the resistance of extremophiles.

**Table 1 microorganisms-10-00610-t001:** Conventional methods for heavy metal removal.

Methods	Description
Chemical precipitation	The most common method for heavy metal removal from solutions. The ionic metals are converted to insoluble forms by chemical reactions using precipitating reagents (precipitants) and form metal hydroxides, sulfides, carbonates, and phosphates (insoluble solid particles) that can be simply separated by settling or filtration.
Electrodialysis (ED) and Electrodialysis Reversal (EDR)	ED and EDR are considered electro-membrane separation processes as ion-exchange membranes (IEM) that are used to separate different ions present in solution as it permeates owing to electrical potential difference. ED/EDR has been mainly utilized for advanced water deionization, high-efficiency removal of ions in pure and ultrapure water application as well as brackish water desalination.
Membrane filtration (MF)	MF is capable of removing not only suspended solid and organic components but also inorganic contaminants such as metal ions. A membrane is a selective layer used to make contact between two homogenous phases with a porous or non-porous structure for the removal of pollutants. Based on the various sizes of the particle, it is divided into three types as below:
○ Ultrafiltration (UF)	UF utilizes permeable membrane to separate heavy metals with pore sizes in the range of 0.1–0.001 micron which permeates water and low molecular weight solutes, while retaining the macromolecules, particles, and colloids that are larger in size of 5–20 nm. The removal of Cu (II), Zn (II), Ni (II), and Mn (II) from aqueous solutions is achieved by using ultrafiltration assisted with chitosan-enhanced membrane with a rejection of 95–100% or a copolymer of malic acid and acrylic acid attaining a removal efficiency of 98.8% by forming macromolecular structures with the polymers.
○ Nanofiltration (NF)	NF is a pressure-driven membrane process that lies between ultrafiltration and reverse osmosis. It is able to reject molecular ionic species by making separation of large molecules possible by small pores when they are within the molecular weight range from 300 to 500 Da with a pore diameter of 0.5–1 nm. A current commercial nanofiltration membrane NF270 is used for removing Cd (II), Mn (II), and Pb (II) with an efficiency of 99, 89, and 74%, respectively.
○ Reverse Osmosis (RO)	In RO, a pressure-driven membrane process, water can pass through the membrane, while the heavy metal is retained. The removal performance of an ultra-low-pressure reverse osmosis membrane (ULPROM) was investigated for the separation of Cu(II) and Ni(II) ions from both synthetic and real plating wastewater.
Microfiltration (MF)	MF uses the same principle as ultrafiltration. The major difference between the two processes is that the solutes which are removed by MF are larger than those rejected by UF using the pore size of 0.1–10 μm with applied pressure range of 0.1–3 bar.
Photocatalysis	Photocatalysis is based on the reactive properties of electron- hole pairs generated in the semiconductor particles under illumination by light of energy. Metal ions are reduced by capturing the photo-excited conduction band electrons, and water or other organics are oxidized by the balance band holes. Heavy toxic metal ions such as Hg^2+^ and Ag^+^, and noble metals can be removed from water by photo deposition on Titania surface-trapped photoelectron states, probably Ti(III), and silver deposition could be observed on the same time scale.

**Table 2 microorganisms-10-00610-t002:** The promising bacterial strains that can remove HMs via biosorption process.

Bacterial Biosorbents	Target Metals	Amount of Heavy Metals Uptake (mg/L)	Biosorption Efficiency (%)	Reference
*Pseudomonas alcaliphila* NEWG-2	Cr	200	96.6	[52]
*Pseudomonas* sp. *strain* DC-B3	Cr	55.35	41	[53]
*Pseudomonas aeruginosa* G12	Cr	10	93	[54]
*Cellulosimicrobium funkei AR6*	Cr	164.66	82.33	[55]
*Stenotrophomonas maltophilia*	Cr	19.84	99.2	[56]
*Acinetobacter* sp. WB-1	Cr	6.82	68.17	[57]
*Cellulosimicrobium* sp.	Cr	96.98	96.98	[58]
*Stenotrophomonas* sp.	Cr	270	90	[59]
*Cellulosimicrobium* sp.	Pb	200	84.62	[58]
*Methylobacterium* sp.	Pb	300	62.28	[60,61]
*Aeribacillus pallidus* MRP280	Pb	86.47	96.78	[62]
*Bacillus* sp. PZ-1	Pb	400	>90	[63]
*Arthrobacter viscosus*	Pb	100	97	[64]
*Arthrobacter* sp. 25	Pb	95.04	86.25	[65]
*Pseudomonas* sp. I3	Pb	49.48	98.96	[66]
*Bacillus badius* AK	Pb	60	60	[67]
*Klebsiellap enumoniae*	Cd	40.18	40.18	[68]
*Rhodotorula* sp.	Cd	40	80	[69]
*Bacillus megaterium* sp.	Cd	39.5	79	[69]
*Bacillus* sp. Q3	Cd	108.2	93.76	[70]
*Pseudomonas aeruginosa* FZ-2	Hg	10	99.7	[71]
*Vibrio parahaemolyticus* PG02	Hg	5	90	[72]
*Pseudomonas aeruginosa*	Cd, Pb	62.8 (Cd); 73.1 (Pb)	87 (Cd); 98.5 (Pb)	[73]
*Saccharomyces cerevisiae*	Pb, Cd	0.045 (Pb); 0.47 (Cd)	70.3 (Pb); 76.2 (Cd)	[74]
*Desulfovibrio desulfuricans* (immobilize on zeolite)	Zn	174	100	[75]
*Micrococcus luteus* DE2008	Pb, Cu	20.4 (Cu); 98.25 (Pb)	25.42 (Cu); 36.07 (Pb)	[76]
*Bacillus* sp.	Pb, Cu, Cd	990 (Cd); 970 (Cu); 200 (Pb)	>90 (Cd, Cu); 20 (Cd)	[77]
*Oceanbacillus profundus*	Pb, Zn	45 (Pb); 1.08 (Zn)	97 (Pb); 54 (Zn)	[78]
*Staphylococcus epidermidis*	Cr, Zn	118 (Zn); 112 (Cr)	59 (Zn), 56 (Cr)	[79]
*Streptomyces* sp.	Pb, Cd, Cu	1.43 (Cu); 0.91 (Pb) 3.66 (Cd)	Pb (83.4); Cu (74.5); Cd (68.4)	[80,81]
*Klebsiella* sp. USL2S	Hg, Pb, Cd, Ni	8500 (Hg); 10,000 (Pb); 1026 (Cd); 8479 (Ni)	85 (Hg); 97.13 (Pb); 73.33 (Cd) 86.06 (Ni)	[82]
*Pseudomonas azotoformans* JAW1	Cd, Cu, Pb	24.64 (Cd); 17.44 (Cu); 19.55 (Pb)	98.57 (Cd); 69.76 (Cu); 78.23 (Pb)	[83]

**Table 3 microorganisms-10-00610-t003:** The potential bacterial strains that can remove HMs via bioaccumulation process.

Bacterial Biosorbents	Target Metals	Amount of Heavy Metals Uptake (mg/L)	Bioaccumulation Efficiency (%)	Reference
*Bacillus megaterium* sp.	Pb	2.1	98.5	[88]
*Bacillus* sp. PZ-1	Pb	100	96	[63]
*Arthrobacter viscosus*	Pb	100	96	[64]
*Bacillus thuringiensis* PW-05	Hg	50	91	[89]
*Vibrio fluvialis*	Hg	0.25	60	[90]
*Enterdobacter cloacae*	Cd	4	72.11	[91]
*Bacillus* sp. Q5	Cd	97.35	76.42	[70]
*Burkholderia cepacia* GYP1	Cd	>90	>90	[92]
*Bacillus cereeus*	Cr	1500	81	[93]
*Sporosarcina saromensis* M52	Cr	50	82.5	[94]
*Pseudomonas aeruginosa* RW9	Cr	0.46	90	[95]
*Rhizopus stolonifer*	Pb, Ni, Cd	170.7 (Pb); 18.7 (Ni); 25.6 (Cd)	44.44 (Pb); 16.66 (Ni); 8.3 (Cd)	[96]
*Bacillus subtilis*	Pb, Cd	2.09 (Pb); 0.37 (Cd)	98.1 (Pb); 92.5 (Cd)	[88]
*Pseudomonas* sp.	Pb, Cd	98.2 (Pb); 82.6 (Cd)	>98 (Pb); 75 (Cd)	[97]
*Streptomyces* K11	Zn	11.76	36	[98]
*Streptomyces zinciresistens*	Cd, Zn	220.5 (Cd); 113.5 (Zn)	98.11 (Cd); 87.33 (Zn)	[99]
*Alcaligenes* sp. MMA	Cr, Zn, Cd	9.78 (Cr); 14 (Zn); 12.6 (Cd)	48.93 (Cr); 70 (Zn); 63 (Cd)	[100]
*Bacillus cereus* RC-1	Zn, Cd, Pb	3.83 (Zn); 8.14 (Cd); 4.03 (Pb)	38.3 (Zn); 81.4 (Cd); 40.3 (Pb)	[101]

**Table 4 microorganisms-10-00610-t004:** Differences between the biosorption and bioaccumulation process of heavy metal removal conducted by bacteria.

Contents	Biosorption	Bioaccumulation
General features	Passive process	Active process
	Ions bound on the surface of ions	Intracellular accumulation of ions
	Rapid and simple process	Requires longer time and complex process
	Not energy requirement	Requires energy sources for metabolisms
	Carried out by both-live and dead biomass	Carried out only by live biomass
	No sensitivity to cultivation conditions	Inhibited by the lack of nutrients, low temperature, and metal toxicity
	Fresh cultivation medium is not necessary	Need of fresh cultivation medium
	Biomass can be regenerated and reuse	Due to the intercellular accumulation, reuse is limited for further purpose
Main affect factors		
pH and temperature	Can occur in a wide range of pH and temperature	Be sensitive to pH and temperature change led to a significant change in living cells
Selectivity	Can be increased by modification or biomass transformation	Better in the case of biosorption
Concentration and type of pollutant	There is a limitation for maximum biosorption	More significant affect cell growth led to more affect the accumulation ability

**Table 5 microorganisms-10-00610-t005:** Major advantages and disadvantages of bacterial biosorbent.

Advantages	Disadvantages
Cost-effective and simple operation owing to utilization of bacterial biomass. Multiple heavy metals uptake at a time. No additional nutrient requirement. Capable of treatment the large volumes of wastewater. Efficient in a wide range of conditions including temperature, pH, salinity, and the presence of various kinds of contaminants. High efficiency by decreasing the volume of solid waste and concentration of pollutants from wastewater. Regeneration of biosorbents.	Saturation of active sites of metal binding ligands. Incomplete metal removal in real conditions. May need the high energy requirements. Living cells are more efficient than dead cells in removal but: ○ It takes a long time to find the bacterial materials. ○ There are difficulties in controlling and managing bacterial growth and activities. ○ The cost of production and maintenance of living biomass may be high.

## Data Availability

The data used to support the findings of this study are available from the corresponding author upon request.

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
