# Peer review of "Bacterial Biosorbents, an Efficient Heavy Metals Green Clean-Up Strategy: Prospects, Challenges, and Opportunities"

_microorganisms, 2022, doi:10.3390/microorganisms10030610_

Round 1

Reviewer 1 Report

  1. Some suggestions to better present the data. Would it be possible to add in the table with examples of microbial biomass (Table 3) values for efficiency?

  2. Section 2.2 presents bioaccumulation, however in Figure 2 are presented Factors affecting the biosorption process. Some discutions need to be added: What parameters affect differently the biosorption and bioaccumulation process? What are gaps of each method? Which method is best? What are the authors recommendations?

  3. The experimental conditions and biosorbent performance (capacity), mechanism of adsorption, regeneration potential should be portrayed in detail.

  4. Future recommendations are a critical part of the review. It should be provided.
  5. In figure 2 - substrate and Aeration conditions  have the same examples: Types of carbon, nitrogen sources/nutrients - not really related to Aeration
  6. Explain the reason for choosing to discuss especially extremophiles in heavy metal removing. 
  7. Could this process be used in industry? What are the reasons for this?
  8. The title refers to Microbial Biosorbents, but only Bacteria has been analysed, with no information on fungi or algae. Either change the title to Bacterial Biosorbents, or add information about other microorganism.

Author Response

Dear Reviewer,

We would like thank you for your valuable time and your helpful comments to make our manuscript better before it can be accepted for the publication. Please kindly find our response in the attached file.

Best regards,

Woojin Chung

Reviewer 2 Report

Dear Mr. Anthony Li

Assistant Editor Microorganisms -MDPI

I read the manuscript microorganisms-1620176-peer-review-v1: “ Microbial Biosorbents, an Efficient Heavy Metals Green 2 Clean-up Strategy: Prospects, Challenges, and Opportunities”, and I present below a few of my observations:

This is a very interesting, useful and well-organized review article.

Be careful with the terms used: microorganisms instead of microbes, throughout the entire manuscript.

I suggest minor corrections before they can be accepted for publication !!!

Sincerely yours,

Prof.dr.Daniela Suteu

Author Response

(The authors gave the same response as above.)

Reviewer 3 Report

Manuscript entitled “Bacterial biosorbents, an efficient heavy metals green clean-up strategy: Prospects, challenges, and opportunities” submitted by Van Hong Thi Pham, Jaisoo Kim, Soonwoong Chang, and Woojin Chung, can be considered for publication in Microorganisms Journal, after a major revision.

Here is a list of my specific comments:

  1. General comment: The utility of this study should be clearly highlighted in the manuscript.
  2. Page 3, lines 72-87: The paragraphs: “Therefore, the development of remediation treatment… both aerobic and anaerobic conditions [27].” should be moved immediately after Table 1.
  3. Page 3, line 77: “However, in such cases, the speed of…”. Add here as reference the paper: doi.org/10.1016/j.jenvman.2018.07.066, because is relevant for this observation.
  4. Page 3, 2. Microbial remediation: the mechanisms of biosorption and bioaccumulation using bacterial biomass as a tool in polluted environmental cleanup: The aspects included in this section should be more detailed discussed. Relevant examples should be added to highlight the importance of this study.
  5. Page 3, line 105: “The bioremediation process takes place…”. Add here some references.
  6. Page 4, Fig. 1: This figure is not visible.
  7. Page 5, 2.1. Biosorption process: Include in this section a table with the most relevant examples.
  8. Page 5, 2.2. Bioaccumulation process: The same observation.
  9. Page 7, Figure 2: This figure is not visible.
  10. Page 8, Table 4: this table should be divided in two parts: one for biosorption and another for bioaccumulation.
  11. Page 8, 3. Potential of extremophiles in heavy metal removing: The aspects included in this section should be more detailed discussed. Relevant examples should be added to highlight the importance of this study.
  12. Page 9, Figure 3: This figure is not visible.
  13. Page 10, Figure 4: The same observation.
  14. Page 10, 4. Concluding remarks and future prospects: Divide this section in two parts: Future prospects and Conclusion. Each of this section should be clearly presented.

Author Response

Dear Reviewer,

We would like to thank you very much for your time and your kind help in our submission. We have revised the MS as you suggested. Please also find our response in the attachment.

Round 2

Reviewer 1 Report

The Figures are not visible in the pdf document. 

Author Response

Dear Reviewer,

We sincerely apologize for the mistake in the previous PDF version. We have submitted the revision in both MS and PDF.

We hope that you will satisfy with our revision. Thank you very much for all your time and your kind help in our submission process in Microorganisms Journal.

All the best,

Woojin Chung

Reviewer 2 Report

Dear Mr. Anthony Li

Assistant Editor Microorganisms -MDPI

I read the manuscript microorganisms-1620176-peer-review-v2: “ Microbial Biosorbents, an Efficient Heavy Metals Green 2 Clean-up Strategy: Prospects, Challenges, and Opportunities”, and I present below a few of my observations:

Although I suggested that the authors change the name "microbe" to "microorganism", they did not want to do so. "Microb" is a more scholastic than scientific terminology. If the Academic Editor considers that keeping this terminology is beneficial for the journal, I agree with the publication of the article.

Sincerely yours,

Prof.Dr.Eng. Daniela Suteu

Author Response

Dear Prof. Eng. Daniela Suteu,

We sincerely apologize for our misunderstanding your mind of the previous time. We have changed them to "bacteria" and "microorganisms". We hope that you will agree with us on this point of view.

Once again, we would like to thank you very much for all your kind help on our submission.

Best regards,

Woojin Chung

Reviewer 3 Report

All my previous remarks and comments have been considered in this new version of the manuscript. In my opinion, the revised manuscript meets the criteria and can be published as review paper in Microorganisms Journal.